# Prediction Models for Brain Distribution of Drugs Based on Biomimetic Chromatographic Data

**DOI:** 10.3390/molecules27123668

**Published:** 2022-06-07

**Authors:** Theodosia Vallianatou, Fotios Tsopelas, Anna Tsantili-Kakoulidou

**Affiliations:** 1Medical Mass Spectrometry Imaging, Department of Pharmaceutical Biosciences, Uppsala University, 751 24 Uppsala, Sweden; 2Laboratory of Inorganic and Analytical Chemistry, School of Chemical Engineering, National Technical University of Athens, 157 80 Athens, Greece; ftsop@central.ntua.gr; 3Faculty of Pharmacy, National and Kapodistrian University of Athens, 157 71 Athens, Greece

**Keywords:** biomimetic chromatography, brain disposition, prediction models, brain BBB fraction unbound, brain unbound volume of distribution

## Abstract

The development of high-throughput approaches for the valid estimation of brain disposition is of great importance in the early drug screening of drug candidates. However, the complexity of brain tissue, which is protected by a unique vasculature formation called the blood–brain barrier (BBB), complicates the development of robust in silico models. In addition, most computational approaches focus only on brain permeability data without considering the crucial factors of plasma and tissue binding. In the present study, we combined experimental data obtained by HPLC using three biomimetic columns, i.e., immobilized artificial membranes, human serum albumin, and α_1_-acid glycoprotein, with molecular descriptors to model brain disposition of drugs. K_p,uu,brain_, as the ratio between the unbound drug concentration in the brain interstitial fluid to the corresponding plasma concentration, brain permeability, the unbound fraction in the brain, and the brain unbound volume of distribution, was collected from literature. Given the complexity of the investigated biological processes, the extracted models displayed high statistical quality (R^2^ > 0.6), while in the case of the brain fraction unbound, the models showed excellent performance (R^2^ > 0.9). All models were thoroughly validated, and their applicability domain was estimated. Our approach highlighted the importance of phospholipid, as well as tissue and protein, binding in balance with BBB permeability in brain disposition and suggests biomimetic chromatography as a rapid and simple technique to construct models with experimental evidence for the early evaluation of CNS drug candidates.

## 1. Introduction

The brain disposition of drug candidates has been one of the major issues in the pharmaceutical industry in recent decades. Τhe development of novel compounds targeting the central nervous system (CNS) is becoming more and more essential as a consequence of the increasing incidence of neurological and neurodegenerative disorders (e.g., depression, Parkinson disease, and Alzheimer disease). The high attrition rate and failures in the field of CNS agents are current challenges that pharmaceutical companies have to face [1,2]. From this perspective, assessment of brain distribution in the early stages of the drug development process is crucial for novel CNS drug candidates. The most important obstacle for drug delivery in the brain is the blood-brain barrier (BBB), which is formed by the endothelial cells lining the brain micro-vessels [3,4,5]. The presence of tight junctions between the endothelial cells leads to limited fenestration, making passive diffusion the dominant transport pathway inside the brain, while the paracellular path is of much lower importance. Moreover, the brain transport of certain drugs may be facilitated or restricted by the presence of membrane transport proteins in the BBB [6].

The most common approach to quantifying the BBB permeability of molecules has been the determination of the ratio between the total brain and the total plasma concentration, mostly expressed as K_p,brain_. However, the reliability of K_p,brain_ in quantitative estimations of brain disposition has been criticized, as its determination is based on the total concentrations, ignoring issues like plasma protein binding (PPB) and tissue binding. According to the “free drug hypothesis”, only the unbound drug crosses the biological membranes, thus constituting the unbound concentration (C_u_) relevant for pharmacological activity. From this perspective, the ratio between the unbound drug concentration in the interstitial fluid of the brain to the corresponding plasma concentration, K_p,uu,brain_, is considered as a more representative measure [7,8,9,10,11,12,13].

Experimental determination of K_p,uu,brain_ through microdialysis exhibits certain limitations as it is technically demanding, time-consuming, and inefficient for very lipophilic compounds [7]. However, K_p,uu,brain_ can be derived from K_p,brain_ by combining the unbound fraction of the drug in the plasma (f_u,p_) and in the brain (f_u,brain_) or the unbound brain volume of distribution, V_u,brain_ [7]. The brain homogenate method is used to estimate the f_u,brain_, considered to reflect only nonspecific binding, while the brain slice method is applied for measuring the V_u,brain_ (mL/g brain), which quantifies the overall cellular uptake of the drug, including active cellular membrane transport and pH partitioning [7,10,12,14]. The relevant equations are included in Section 3. Evidently, f_u,p_ is obtained by standard methods for plasma protein binding.

K_p,brain_, usually converted into the logarithmic form and labeled as logBB, has been widely used to establish in silico models in an effort to reveal the favorable physicochemical and molecular properties for BBB permeability [15]. A number of QSAR models have been reported in the literature, most of them involving calculated lipophilicity, expressed as octanol–water partition or distribution coefficients (logP or logD), polar surface area, and molecular weight [16,17,18,19]. However, to the best of our knowledge, analogous models for f_u,brain_, V_u,brain_, or K_p,uu,brain_ itself are rather limited [11,14,20].

A potential concern regarding in silico predictive models may arise in respect to the reliability of calculated physicochemical properties of drugs, in particular for new chemotypes. It has been reported that small errors in logP or logD predictions may cause substantial errors in the estimation of biological end-points, particularly if limited datasets are analyzed, while experimental lipophilicity requires a rather tedious and time-consuming procedure [21,22]. On the other hand, user-friendly chromatographic techniques may offer a challenging alternative to combining rapid measurements with theoretical descriptors for the construction of ‘hybrid’ in silico models based on experimental evidence.

Biomimetic properties, defined as the retention outcome on HPLC stationary phases containing a biologically relevant agent, have attracted considerable interest and are currently used for the rapid evaluation of ADME properties in early drug discovery phases. In particular, immobilized artificial membrane (IAM) chromatography, which uses phospholipid-containing stationary phases, has been applied in investigating permeability as an alternative to traditional octanol–water lipophilicity. Since, however, electrostatic interactions have a strong contribution in retention mechanism, especially in the case of protonated bases, IAM chromatography is considered as also reflecting drug–membrane interactions and tissue binding [23,24]. IAM models for oral absorption, skin partitioning, and brain penetration, mostly expressed as logBB, have been reported in the literature, usually in combination with additional molecular descriptors [23,24,25,26,27,28,29,30,31,32,33,34,35,36,37,38,39,40,41,42]. Classification of CNS^+^ and CNS^−^ drugs has been suggested using the IAM retention factor divided by the fourth root of molecular weight [43]. On the other hand, protein-based stationary phases, incorporating human serum albumin (HSA) or α_1_ acid glycoprotein (AGP)-, simulate binding to plasma proteins [25,26,39,44,45,46].

Immobilized HSA retains the characteristics of the protein in solution, as has been proven by zone and frontal analysis experiments, permitting the safe estimation of binding constants to human serum albumin in the plasma [47]. Less investigated is AGP retention due to the polymorphism of α_1_ acid glycoprotein and practical issues related to the immobilization on the silica skeleton of the column. It is well-established, however, that protonated bases are more strongly retained on AGP columns, in agreement with the same binding preference of AGP in solution [47]. In light of the above considerations, HPLC-based biomimetic properties reflect the major factors which govern most biological processes, e.g., passive diffusion and non-specific binding to phospholipids, tissue, or plasma proteins. Accordingly, their combination may be suitable for the estimation of composite pharmacokinetic data. Valko et al. used the weighted sum of IAM and HSA retention to estimate the volume of distribution and tissue binding in different organs, including brain tissue binding [39].

In the present study, we investigated the performance of biomimetic properties in combination with molecular descriptors in the development of ‘hybrid’ models with experimental evidence for the thorough analysis of composite K_p,uu,brain_ data, and the distinct brain disposition components, K_p,brain_, f_u,brain_, and V_u,brain_. The aim was to bridge the gap with pure in silico prediction models and to contribute to the perception of these crucial experimental end-points for CNS-acting drug candidates. For this purpose, in-house retention factors, determined for a number of pharmaceutical compounds on IAM, has, and AGP stationary phases, were used together with physicochemical/molecular descriptors. Lipophilicity, expressed as octanol–water partition coefficients (logP) or distribution coefficients at pH 7.4 (logD_7.4_) and pH 5.0 (logD_5.0_), was incorporated in the pool of descriptors, and its implementation in the models was compared to that of IAM retention. Models were constructed by applying multiple linear regression (MLR) and partial least squares (PLS) analysis. MLR can provide simple models that are easily used by medicinal chemists, while PLS, contributing complementary and supporting MLR models, may permit a deeper insight into the factors underlying brain disposition of drugs. Attention was given to model validation in respect to robustness and applicability domain to offset the drawback that rather limited datasets were analyzed.

## 2. Results and Discussion

In the present study, we collected experimental biomimetic chromatographic data, previously obtained in our laboratory, for a set of 55 pharmaceutical compounds belonging to a wide range of pharmacological classes. (For further details, see the Section 3 and Appendix A.) The investigated compounds are small molecules within a molecular weight (MW) range of ca 129 to 513 Da (Figure 1a). The dataset includes neutral, basic, and acidic compounds; however, most molecules contain one basic ionizable group (Figure 1b).

Experimentally measured logBB, K_p,uu,brain_, f_u,brain_, and V_u,brain_ values were collected from the literature [10,11,12,48,49,50,51,52,53,54] (Appendix A). The collected logBB values indicated a sufficient range of BBB permeability, from highly permeable (logBB > > 1) to significantly less permeable (logBB < < 0) compounds (Figure 1c). Nevertheless, the range of logK_p,uu,brain_ values suggested a lower extent of brain distribution of the studied molecules (Figure 1e). The two measures showed a mediocre intercorrelation (Figure 1e), while f_u,brain_ and V_u,brain_ showed significant inverse intercorrelation on the basis of their logarithmic values (Figure 1f–h).

For further statistical analysis, f_u,brain_ values were converted to the thermodynamic constant K_b_ [55] according to the relevant equation included in Section 3. Its logarithmic form logK_b_ was used throughout.

### 2.1. Data Overview with Unsupervised Data Analysis

To obtain an overview of the dataset, unsupervised principal component analysis (PCA) was performed using the pool of descriptors and the response variables as an X matrix. A 5-component PCA model was generated with R^2^ = 0.674 and Q^2^ = 0.462. The score plot of the first two components shows a uniform distribution of the data in all four quartiles, with one drug, candesartan, lying outside the Hoteling T^2^ ellipse (Figure 2a). As depicted by the coloring of the scores based on the logk_wIAM_ values, IAM retention is a dominant parameter of the data distribution. The loadings plot reflects the correlations among the chromatographic data, the molecular and physicochemical descriptors, and the experimental brain distribution data in a multilayered fashion (Figure 2b). The coloring of the variables corresponds to their grouping according to hierarchical clustering based on the five principal components. LogBB and logV_u,brain_ are located in the same quartile with chromatographic factors and lipophilicity parameters and belong to the same hierarchical clustering group. Interestingly, logK_p,uu,brain_ is also incorporated into the same group, but its loading value in the first component is approximately zero. LogK_b_ is located in the opposite quartile in high proximity to hydrogen bonding parameters.

### 2.2. Modeling logK_p,uu,brain_

LogK_p,uu,brain_ is considered to be the most relevant measure of the rate and extent of drug delivery in the brain, and as such, it was examined first. MLR analysis gave poor statistics (R < 0.5). Polarity descriptors, e.g., tPSA, display the strongest negative correlation (Figure 3a). Poor correlation was observed between logK_p_,_uu,brain_ and logk_wIAM_, logP or logD7.4. However, a relatively better correlation was found between logD_5.0_ and IAM retention at pH 5.0 (Figure 3a).

The above findings were further supported by Partial Least Squares analysis. PLS analysis offers a number of advantages, tolerating intercorrelated variables and missing values. Variables are treated simultaneously to extract the principal components as their linear combinations. Using the entire pool of descriptors and performing variable selection according to the variable importance to projection (VIP) criterion, no satisfactory PLS models could be obtained, with R^2^ and Q^2^ not exceeding 0.52 and 0.32, respectively. In agreement with MLR, polarity terms (TPSA) show the strongest impact with VIP values > 1 and negative contribution. Biomimetic properties and lipophilicity have positive contributions, albeit with considerably lower impacts. In agreement with MLR results, logD_5.0_ and IAM retention at pH 5.0 show higher influence. In fact, VIP values decrease in the order of: logD_5.0_ > logk_wIAM,5.0_ > logD_7.4_ > logk_10,HSA_ > logk_wAGP_ > logP > logk_wIAM,7.4_ (Appendix A). The better performance of IAM retention and lipophilicity at pH 5.0 may be explained by the recently reported evidence of the interference of the endo-lysosomal system (lysosomes’ pH 5.0–4.5) in vitro blood–brain barrier models [56].

Inspection of the plots of observed vs. predicted logK_p,uu,brain_ and of DModY plots (distance of observations to the Y model) revealed two strong outliers, quinidine and theophylline, present in any PLS model (figure not shown). Upon exclusion of these drugs from statistical analysis, acceptable PLS models were obtained. Using biomimetic properties and lipophilicity in combination with computational descriptors, a 1-component PLS model with 54 original descriptors was generated with the following statistics (PLS Model 1):A = 1, n = 23, R^2^ = 0.694 Q^2^ = 0.518, RMSEE = 0.262 and RMSEcv = 0.315. (PLS Model 1)

The separate effect of IAM retention or lipophilicity is shown in PLS Models 2 and 3, which include either IAM retention at pH 7.4 and 5.0 or lipophilicity measures (logP, logD_7.4_ and logD_5.0_). PLS Models 2 and 3 denote equal performance of IAM retention and traditional lipophilicity:A = 1, n = 23, R^2^ = 0.693 Q^2^ = 0.529, RMSEE = 0.263 and RMSEcv = 0.311 (PLS Model 2)
A = 1, n = 23, R^2^ = 0.694 Q^2^ = 0.529, RMSEE = 0.263 and RMSEcv = 0.311 (PLS Model 3)

Finally, the use of only a few computational descriptors with high VIP values, including polarity, flexibility, and shape indices, proved sufficient to produce an improved, 1-component PLS model (PLS Model 4):A = 1, n = 23, R^2^ = 0.686 Q^2^ = 0.621, RMSEE = 0.266 and RMSEcv = 0.279 (PLS Model 4)

In Figure 3b,c the plot of observed vs. predicted logK_p_,_uu,brain_ values generated by PLS Model 4 and the coefficients of the original variables are depicted, respectively. The predicted logK_p_,_uu,brain_ values are presented in Appendix A. The applicability domain was defined using DModY plot, and the doubled average DModY value, equal to 0.822 × 2 = 1.644, was used as a critical value. Two drugs, acetaminophen and midazolam, exceed the critical value (Appendix A).

PLS Model 4 was validated by permutation tests (Appendix A) and by dividing the dataset into a training set and a test set, using 4 different training/test sets (see Section 3). The external validation demonstrated robust statistics except for 1 iteration, in which the drug acyclovir was included in the test set (Table 1). In Appendix A, a representative plot of observed vs. predicted values is presented.

In fact, the construction of PLS Model 4 indicates that biomimetic properties and lipophilicity are less crucial in logK_p_,_uu,brain_ modeling, as shown also by the failure to generate successful MLR models. Previous studies have likewise identified the lack of strong correlations between logK_p,uu,brain_ and physicochemical descriptors, especially lipophilicity, while hydrogen bonding has been found to be the strongest contributor [11,20]. Loryan, et al. reported a PLS model with two descriptors related to polarity (tPSA) and hydrogen bonding capacity [14].

Considering the complex nature of K_p,uu,brain_ as the outcome of opposing brain disposition end-points and plasma binding [11] (see relevant equations in Section 3), modeling K_p,brain_*,* expressed as logBB, f_u,brain_, and V_u,brain_, can indirectly assist in the exploration of logK_p,uu,brain_. In the next sections, the generation and validation of the MLR and PLS models of these distinct brain disposition end-points are discussed.

### 2.3. Modeling logBB

#### 2.3.1. Multiple Linear Regression Models

Partial correlations between logBB, IAM retention, logD_7.4_, and physicochemical/molecular properties revealed the crucial effect of polarity, with TPSA showing the highest correlation coefficient (r = −0.777, *p* < 0.001), followed by the sum of nitrogen and oxygen atoms, NO (r = −0.711, *p* < 0.001), and dipole moment (r = −0.693, *p* < 0.001), while for logkwIAM and logD7.4 r = 0.617 (*p* < 0.001) and r = 0.687 (*p* < 0.001), respectively. Their combination with TPSA led to Equations (1) and (2), respectively. The term h* in Equations (1) and (2) expresses the critical leverage value, which defines the applicability domain (AD) [57]. (See Section 3.)
logBB = 0.16(±0.07) logk_wIAM_ − 0.02(±0.003) TPSA +0.69(±0.27)
n = 41*, R = 0.806, R^2^ = 0.650, R^2^adj = 0.632, s = 0.472, F = 35.31, h* = 0.073(* Piracetam did not have a logk_wIAM_ measured value).(1)

Eight drugs had h values higher than the threshold h* value and were beyond the AD of the model, as also shown in a Williams plot (Appendix A).
logBB = 0.15(±0.05) logD_7.4_ − 0.02(±0.003) TPSA +0.72(±0.23)
n = 42, R = 0.816, R^2^ = 0.665, R^2^adj = 0.648, s = 0.456, F = 38.74, h* = 0.071(2)

Six drugs were outside the AD of Equation (2) (Appendix A).

In both equations, TPSA had a higher significance than logk_wIAM_ or logD_7.4_, with Student’s test-t values of |t_TPSA_| = 5.4 and 4.8, respectively, while |t_logkwIAM_| = 2.22, and |t_logD7.4_| = 2.69.

External validation of Equations (1) and (2) was performed by dividing the dataset into a training set and a test set. This procedure was repeated up to six times for each equation. Practically speaking, the same equations were generated with comparable statistical data (Table 2). In all cases, observed vs. predicted values showed a 1:1 correlation. In Appendix A, a representative plot of observed vs. predicted values from Equations (1) and (2) is illustrated.

#### 2.3.2. Partial Least Squares Models

Initially, PLS analysis was performed considering both biomimetic properties and lipophilicity in the pool of descriptors, so as to rank their impact in modeling logBB. Polarity descriptors (TPSA and dipole moment) were found to have the highest VIP values (VIP = 1.2). AGP retention showed equally high impact, followed by logD_7.4_ (VIP = 1.05), and IAM retention (VIP = 0.95). HSA retention and logP had a lower influence in the model with VIP < 0.9.

Keeping AGP retention as the most influential parameter and including either logk_wIAM_ (PLS Model 5, Figure 4a,b) or logD_7.4_ (PLS Model 6, Figure 4c,d), 2 component PLS models with 19 descriptors were obtained after variable selection, with practically equal statistical quality:A = 2, n = 42, R2 = 0.846, Q2 = 0.792, RMSEE = 0.313, RMSEcv = 0.351 (PLS Model 5)
A = 2, n = 42, R2 = 0.846, Q2 = 0.791, RMSEE = 0.313, RMSEcv = 0.351 (PLS Model 6)

The applicability domain was defined by the critical value DModY being 0.767 × 2 = 1.534 for Model 5 and 0.789 × 2 = 1.58 for Model 6 (Appendix A). A total of 4 drugs, i.e., atenolol, candesartan, haloperidol, and maprotiline, were outside the AD of PLS Model 5. The same drugs, plus indomethacin, were outside the AD of PLS Model 6 (Appendix A).

Considering the large percentage of missing logk_wAGP_ values, and in order to confirm its essential contribution, PLS analysis was repeated including only drugs with available logk_wAGP_ data. The generated models proved the crucial contribution of AGP retention (Appendix A).

The contribution of logk_wAGP_ in logBB was further confirmed in MLR models for the drugs with available data:logBB = 0.467(±0.182) logk_wAGP_ − 0.013 (±0.05)TPSA + 0.018(±0.483)
n = 19, R = 0.871, R^2^ = 0.758, R^2^ adj = 0.728, s = 0.431, F = 25.1(3)

The high impact of AGP retention may be related to the presence of a strong negative charge in AGP stationary phases due to their high content in sialic acid. Brain cell membranes are also the most anionic and have their lipids mostly exposed, thus explaining the reason that lipophilic cationic compounds are more prone to cross the BBB.

PLS Models 5 and 6 were validated with permutation tests (Appendix A) and by external validation upon dividing the dataset into 5 different training and test sets. Practically speaking, the same models were generated by an external validation procedure, with comparable statistical data (Table 2), with the exception of iteration, including quinidine (PLS Model 5) and morphine (PLS Model 6). In all cases, observed vs. predicted values showed a 1:1 correlation. In Appendix A, a representative plot of observed vs. predicted values is illustrated. The predicted logBB values from the MLR and PLS models are provided in Appendix A.

### 2.4. Modeling logK_b_

#### 2.4.1. Multiple Linear Regression

Since logK_b_ is related to tissue binding, a satisfactory negative correlation with logk_w,IAM_ values was obtained. Strong IAM retention corresponds to high tissue phospholipid binding and thereupon to a lower fraction being unbound:logK_b_ = −0.984(±0.059) logk_w,IAM_ + 0.966(±0.131)
n = 39, R = 0.940, R^2^ = 0.884, R^2^adj = 0.881, s = 0.543, F = 281.786(4)

A negative correlation with inferior but still acceptable statistics was obtained using HSA retention factors:logK_b_ = −1.291(±0.105) logk_10,HSA_ − 0.551(±0.092)
n = 39, R = 0.897, R^2^ = 0.804, R^2^adj = 0.799, s = 0.543, F = 152.245(5)

Further stepwise MLR analysis led to a three-parameter equation, combining the IAM and HSA retention factors with the count of nitrogen and oxygen atoms present in the molecule (NO):logK_b_ = −0.617(±0.093) logk_w,IAM_ − 0.450(±0.126) logk_10,HSA_ + 0.111(±0.034) NO − 0.026(±0.230)
n = 39, R = 0.966, R^2^ = 0.932, R^2^_adj_ = 0.927, s = 0.328, F = 160.776(6)

The positive sign of the coefficient of NO indicated that a higher hydrogen bond acceptor potential is not favorable for unspecific hydrophobic tissue and plasma protein binding, increasing the amount of the unbound fraction. It should be mentioned that logk_w,IAM_ and logk_10ACN,HSA_ showed a considerable degree of intercorrelation (r = 0.824). To overcome the collinearity problem, and considering the low differentiation of regression coefficients of logk_w,IAM_ and logk_10ACN,HSA_ in Equation (5) was within statistical limits, their sum Sum_IAM-HSA_ was used instead:logK_b_ = −0.55(±0.03) Sum_IAM-HSA_ + 0.11(±0.03) NO − 0.14(±0.18)
n = 39, R = 0.965, R^2^ = 0.931, R^2^_adj_ = 0.927, s = 0.327, F = 243.095, h* = 0.231(7)

All drugs were within the AD (Appendix A).

Satisfactory correlation was also obtained with AGP retention factors for 20 drugs with available data:logK_b_ = −1.240(±0.133) logk_w,AGP_ + 1.034(±0.264)
n = 20, R = 0.910, R^2^ = 0.827, R^2^_adj_ = 0.818, s = 0.458, F = 160.863(8)

Hydrophobic binding, a major force in drug-tissue and drug-protein interactions, was better simulated by logP of the neutral form than logD_7.4_. Thus, a satisfactory logK_b_/logP relationship was obtained—although with a lower correlation coefficient, compared to Equation (4):logK_b_ = −0.580(±0.042) logP + 0.614(±0.139)
n = 39, R = 0.915, R^2^ = 0.836, R^2^_adj_ = 0.832, s = 0.497, F = 189.249, h* = 0.153(9)

Further stepwise regression did not lead to an improved equation.

Correlation with the distribution coefficient logD_7.4_ led to an inferior relationship:logK_b_ = −0.555(±0.042) logD_7.4_ − 0.147(±0.142)
n = 39, R = 0.823, R^2^ = 0.677, R^2^_adj_ = 0.668, s = 0.698, F = 77.626(10)

Equation (10), however, can be improved if the fraction of the ionized molecular species F^+^ and F^−^ is introduced in combination with NO. The introduction of fractions F^+^ and F^−^ reflects the binding conditions in respect to basic and acidic drugs, the first interacting stronger with phospholipids and the latter with serum albumin:logK_b_ = −0.54(±0.05) logD_7.4_ − 0.77(±0.18) F^+^ − 0.98(±0.30) F^−^_7.4_ +0.19(±0.05)NO − 0.30(±0.28)
n = 39, R = 0.931, R^2^ = 0.867, R^2^_adj_ = 0.852, s = 0.467, F = 55.57, h* = 0.307(11)

Evidently, all of the regression coefficients in Equation (11) had negative signs, except NO, which contributed positively. Three drugs were outside the AD, as shown in Appendix A.

External validation of Equations (7) and (11) was performed by dividing the dataset into the training set and the test set, using 5 different test sets (Table 3). In all training sets, observed vs. predicted values showed a 1:1 correlation. In Appendix A, a representative plot of observed vs. predicted values from Equations (7) and (11) is illustrated.

#### 2.4.2. Partial Least Squares Models

Application of PLS to model logK_b_ led to a two-component model (PLS Model 7) if both biomimetic properties and lipophilicity were included in the pool of descriptors:n = 39, A = 2, R^2^ = 0.929, Q^2^ = 0.904, RMSEE = 0.332, RMSEEcv = 0.371 (PLS Model 7)

Biomimetic properties and lipophilicity were the most crucial parameters (VIP > 1), with decreasing importance following the order: logk_AGP_ > logk_wIAM_ > logP > logk_10HSA_ > logD_7.4_ (Appendix A). In accordance with the MLR equations, the three biomimetic properties and lipophilicity had negative contributions, while the hydrogen bonding acceptor descriptor NO had a positive effect. More to the point, logP showed a higher impact in respect to logD_7.4_ as a better lipophilicity expression for hydrophobic binding. Three bulk descriptors, CMR, arC6, and nPSA, included in the model had a negative sign (Appendix A).

Including only biomimetic properties in the pool of descriptors, a 3-component model (PLS Model 8) was obtained with improved statistics:A = 3, n = 39, R^2^ = 0.938, Q^2^ = 0.913, RMSEE = 0.315, RMSEEcv = 0.353 (PLS Model 8)

As illustrated in Figure 5, the same signs were kept in PLS Model 8 for biomimetic properties and NO, while a balance was observed between the non-polar descriptors, with nPSA having a positive sign.

According to the DModY criterion (critical value = 0.812 × 2 = 1.624), three drugs, i.e., fluoxetine, ranitidine, and theophylline were outside of the AD of the model (Appendix A).

Replacement of IAM retention by lipophilicity, expressed both as logP and logD_7.4_ led to a 3-component model (PLS Model 9):A = 3, n = 39, R^2^ = 0.925, Q^2^ = 0.844, RMSEE = 0.347, RMSEEcv = 0.478(PLS Model 9)

Five drugs—acetaminophen, atenolol, neostigmine, propranolol, and theophylline—are the beyond AD, with DModY being higher than the critical value (2 × 0.768) (Appendix A).

The logK_b_ PLS Models 8 and 9 were validated with permutation tests (Appendix A), and upon dividing the dataset into 5 different training and test sets. Robust statistical data were obtained in all cases (Table 3). Representative plots of observed vs. predicted logK_b_ values from external validation are presented in Appendix A. The predicted logK_b_ values from the MLR and PLS models are presented in Appendix A.

Back calculation of f_u,brain_ values using the predicted logK_b_ were successfully correlated with the corresponding experimental values, approximating a 1:1 correlation (Appendix A).

### 2.5. Unbound Volume of Distribution, Vu, Brain

#### 2.5.1. Correlation between fraction unbound and unbound volume of distribution in the brain

The negative correlation of logV_u,brain_ with logf_u,brain,_ shown also in Figure 1h, is reflected in Equation (12):logV_u,brain_ = −0.922(±0.091) logf_u,brain_ + 0.386(±0.117)
n = 17, R = 0.934, R^2^ = 0.873, R^2^_adj_ = 0.864, s = 0.319, F = 102.678(12)

Equation (12) is considerably improved upon the inclusion of F^+^, which has a positive effect, indicating the importance of positive charge in the overall cellular uptake (Equation (13):logV_u,brain_ = −0.950(±0.049) logf_u,brain_ + 0.632(±0.102) F^+^_7.4_ − 0.113(±0.102)
n = 17, R = 0.983, R^2^ = 0.966, R^2^_adj_ = 0.961, s = 0.167, F = 199.802(13)

The high quality of Equation (13) permits the safe prediction of logV_u,brain_ from logf_u,brain_.

We further attempted to construct models for the unbound volume of distribution using biomimetic properties, lipophilicity, and computational descriptors although, in this case, the dataset was limited, including only 17 drugs with available experimental data.

#### 2.5.2. Multiple Linear Regression

Direct correlation of logV_u,brain_ with logk_w,IAM_ led to Equation (14), with moderate statistics:logV_u,brain_ = 0.624(±0.093) logk_w,IAM_ + 0.226(±0.190)
n = 17, R = 0.866, R^2^ = 0.750, R^2^_adj_ = 0.734, s = 0.446, F = 45.043(14)

Using stepwise regression, a considerably improved regression equation was obtained upon inclusion of the count of the hydrogen bond acceptors (HBA) as an additional parameter (Equation (15)):logV_u,brain_ = 0.623(±0.071) logk_w,IAM_ − 0.214(±0.062) HBA + 0.958(±0.256)
n = 17, R = 0.930, R^2^ = 0.865, R^2^_adj_ = 0.846, s = 0.339, F = 44.983, h* = 0.53(15)

All compounds were within the AD of Equation (15) (Appendix A).

HSA retention was not significant, while AGP retention led to a very good correlation for the limited dataset of seven compounds:logV_u,brain_ = 0.822(±0.155) logk_wAGP_ + 0.127(±0.280)
n = 7, R = 0.921, R^2^ = 0.849, R^2^_adj_ = 0.819, s = 0.284, F = 28.07(16)

Replacing logk_w,IAM_ with lipophilicity, a moderate regression equation was obtained (Equation (17)), with logP in combination with the fraction protonated, including F^+^ as an additional parameter:logV_u,brain_ = 0.336(±0.054) logP + 0.859(±0. 285) F^+^_7.4_ − 0.115(±0.290)
n = 17, R = 0.864, R^2^ = 0.746, R^2^_adj_ = 0.710, s = 0.465, F = 20.61, h* = 0.53(17)

All compounds were within the AD of Equation (17) (Appendix A).

The combination of logP with the fraction protonated at pH7.4, both with a positive contribution, has previously been reported for a PLS model of the apparent volume of distribution [58].

Owing to the reduced dataset, test sets for the external validation of Equations (15) and (17) contained 4 to 6 compounds. Robust statistical data were obtained for validated equations, except when metformin was included in the test set, for both Equations (15) and (17) (Table 4). In Appendix A, a representative plot of observed vs. predicted values from Equations (15) and (17) is illustrated:

#### 2.5.3. Partial Least Squares (PLS) Models

A one-component PLS model, based on IAM retention, was obtained after variable selection with very good statistics (Figure 6a,b):A = 1, n = 17, R^2^ = 0.932, Q^2^ = 0.901, RMSEE = 0.232, RMSEEcv = 0.264 (PLS Model 10)

In agreement with the MLR models, IAM and AGP retention were the most influential variables with a positive contribution, while HSA retention factors were not included in the final model. Most polar descriptors had a negative effect, and the opposite was true for non-polar descriptors.

According to the DModY criterion (critical value: 0.802 × 2 = 1.604), 2 drugs (metformin and propranolol) were outside the AD of the model (Appendix A).

The use of logP in place of biomimetic properties led to a three-component model with satisfactory statistics although they had inferior cross-validation results (Figure 6c,d):A = 3, n = 17, R^2^ = 0.912, Q^2^ = 0.723, RMSEE = 0.285, RMSEEcv = 0.403 (PLS Model 11)

According to the DModY criterion (critical value 1.524), 2 drugs (pindolol and propranolol) were outside the AD of the model (Appendix A).

The lower Q^2^ and higher RMSEEcv indicate inferior predictability, reflected also in the higher intercepts of the corresponding permutation tests (Appendix A). The PLS models were also validated by external validation, dividing the dataset into two training and test sets. In Table 4, the statistical data of the new models are given. In Appendix A, representative plots of observed vs. predicted logV_u,brain_ are provided. The predicted logV_u,brain_ values from the MLR and PLS models are presented in Appendix A.

Considering the potential of Equation (13) for safe predictions of logV_u,brain_, we used Equation (13) in order to extend the dataset. Drugs with predicted logV_u,brain_ were used as a blind test set. As illustrated in Appendix A, the blind test set was well accommodated in the model with the exception of two drugs, clomipramine and fluoxetine, which were strong outliers. Excluding clomipramine and fluoxetine, the combination of the blind test set with the training set led to PLS Model 12, which was practically the same as PLS Model 10:A = 1, n = 37, R^2^ = 0.858, Q2 = 0.845, RMSEE = 0.355, RMSEEcv = 0.361 (PLS Model 12)

For 4 drugs, DModY exceeded the critical value of 1.45, being outside the AD (Appendix A).

## 3. Materials and Methods

### 3.1. Dataset and Chromatographic Data

A dataset of 55 pharmaceutical compounds, belonging to a wide range of pharmacological classes, was used in the present investigation. The dataset exhibited adequate structural diversity, consisting of acidic, neutral, basic, and zwitterionic molecules. The compounds had been previously studied in our laboratory with respect to their retention profiles in 3 biomimetic chromatographic columns, namely an IAM.PC.DD2 (Regis Technology, Morton Grove, IL, USA), a ChromTech CHIRAL-HSA column (50 mm × 4 mm i.d.), and a ChromTech CHIRAL-AGP column (50 × 4 mm i.d.), under experimental conditions as described in the corresponding references [23,24,27,29,45].

For IAM chromatography, retention factors logk_wIAM_ corresponding to a 100% aqueous mobile phase were used. PBS was used as a buffer at two pH values, i.e., the physiological pH 7.4 (data labeled as logk_wIAM_) and pH 5.0 (data labeled as logk_wIAM5_._0_), with the latter being associated with intestinal absorption and lysosomal trapping [59]. In the case of HSA chromatography, isocratic logk values, measured in the presence of 10% acetonitrile (ACN), were used (logk_10,HSA_), as they showed a highly significant 1:1 correlation with plasma protein binding data in our previous study [45]. Logk_10,HSA_ data were available for 37 compounds in the dataset. For the remaining compounds, logk_10ACN,HSA_ values were calculated based on the highly significant equation reported in the same study [45]. Logk_wAGP_ retention factors corresponding to a 100% aqueous mobile phase were available for 28 drugs. Thus, they could be used in the MLR models only for restricted datasets. However, their performance could be evaluated in PLS models since this type of analysis tolerates missing values. All chromatographic indices are included in Appendix A.

### 3.2. Brain Disposition Data

Experimental K_p,uu,brain_ values measured in rat brain tissue were collected for 22 compounds [10,11,12,60,61] and converted to the logarithmic form. Experimentally determined K_p,brain_ values for 42 compounds were obtained from the literature and were converted to logBB. They preferably referred to rat studies in order to achieve homogeneity in the data [19,31,32]. Experimental f_u,brain_ values were available from the literature for 39 compounds in the dataset [10,50,51,52,53,62]. When more than one value was available, the average was calculated. Most values were determined on rats. Since there are only small inter-species differences, values from species other than rats were included (seven cases) for compounds where rat values were not provided. The f_u,brain_ values were converted to the thermodynamic constant K_b_ according to the following equation [55]:$$\mathrm{Kb}\;=\frac{\mathrm{fu},\mathrm{brain}}{1.001-\mathrm{fu},\mathrm{brain}}$$

The denominator was set to 1.001-f_u,brain_ in the case of compounds exhibiting an f_u,brain_ value equal to unity. The decimal logarithm of K_b_, logK_b_, was used for the development of the models.

V_u,brain_ values, determined using the brain slice method, were collected from the literature for 17 compounds in the dataset and converted to the corresponding logarithm, logV_u,brain_ [10,12].

K_p,brain_, K_p,uu,brain_, f_u,brain_, and V_u,brain_ values are presented in Appendix A. They span a sufficient range and are evenly distributed, including drugs with different brain distribution profiles.

K_p,uu,brain_ can be derived from K_p,brain_ by combining the unbound fractions of the drug in the plasma (f_u,p_) and in the brain (f_u,brain_) or the unbound brain volume of distribution, V_u,brain_ [7]:$$\mathrm{Kp},\mathrm{uu},\mathrm{brain}\sim \frac{\mathrm{Kp},\mathrm{brain}}{\mathrm{Vu},\mathrm{brain}\times \mathrm{fu},\mathrm{plasma}}$$
V_u,brain_ shows an inverse relation to Vu,brain shows an inverse relation to fu,brain approximating under circumstances [63]. fu,brain∼[1/Vu,brain]range 0-1.

### 3.3. Physicochemical and Molecular Descriptors

A pool of 121 descriptors, including 1D, 2D and 3D descriptors, was calculated using appropriate software. The software ADME Boxes 3.0 was used to calculate the topological polar surface area (TPSA), hydrogen bond acceptor and donor sites (HBA and HBD), the rotatable bonds (RB), the number of ionizable groups, and the fraction of molecular species at pH 7.4 (with the fraction of negatively charged species being F^−^_7.4_ and that of positively charged species being F^+^_7.4_). The Abraham’s solvation parameters (hydrogen bond acidity (A), hydrogen bond basicity (B), excess molar refraction (E), McGowan’s volume (V), and dipolarity/polarizability (S)) were calculated with the module ABSOLV implemented using the same software. Topological indices and electrotopological state indices [64] were computed with Molconn-Z (v4.12 eduSoft, LC, La Jolla, CA 92037 USA) software Hyperchem v.5.0/Chemplus v.1.6 software (HYPERCUBE Inc., Waterloo, ON, Canada) was used for the calculation of 3D descriptors. Molecular size descriptors, energy parameters, and dipole moment were calculated using the lowest energetic conformation. Descriptors based on essential structural characteristics, such as the total number of rings (nRings), the number of phenyl rings, the number of heteroatoms inside the rings (Nr, Or, Sr), the number of heteroatoms outside of the rings (Nnr, Onr, and Snr), or the sum of them (N, O, and S), the number of different halogens (Cl, F, I, and Br) and the sum of them (halogen), and the number of double bonds between carbon atoms were derived manually.

Experimental lipophilicity data, logP, logD_7.4_, and logD_5.0_, were taken from references [23,27,29] and the therein-cited literature sources. They are included in Appendix A.

### 3.4. Statistical Analysis

Multiple linear regression analysis (MLR) was performed using SPSS v.22.0 software. Variable selection was performed by applying the stepwise algorithm, first in groups of descriptors, classified according to their physicochemical content, in order to exclude the less-relevant variables for the final selection. Variables with zero or very small variance were excluded, as were collinear variables, considering a correlation coefficient lower than 0.8. The models were evaluated by considering the values of R, R^2^, s (standard deviation), and F- test. For the significance of each individual variable, a t-test value of |t| ≥ 2 was considered. The applicability domain (AD) of the regression equations was defined by calculating the critical leverage h* according to the formula h* = 3 (*p* + 1)/n, where *p* is the number of parameters, and n is the number of compounds [65].

Models were validated by dividing the dataset into different training and test sets of 5 to 9 drugs, depending on the overall sample size. The test sets were randomly selected, taking however into account to cover all four quartiles of the PCA scores plot. Subsequently, each model derived by the training set was used to predict the response variable of the corresponding test set. The R, R^2^, and s for both the training and test sets were considered and compared to the original model. In addition, models were evaluated by the proximity of the relation of observed vs. predicted values to a 1:1 correlation, reflected by a slope close to 1 and an intercept close to 0.

Multivariate data analysis was performed using Simca-P 14.1 (Sartorius Stedim Biotech, Umeå, Sweden,). Prior to analysis, data were scaled to unit variance. Principal component analysis (PCA) was performed, considering all columns of the table as X variables. PCA is a projection method resulting in dimensional reduction. The principal components, derived through projection, represent the new (latent) variables that summarize the information included in the initial set of descriptors. A PCA scores plot provides a useful data overview, which was also considered for the test set selection. Partial least squares analysis (PLS), a regression extension of PCA, was applied in order to construct prediction models. The variable selection was based on the values of the variable’s influence to projection (VIP), the weight (w) in the loadings plot, and the size of coefficients. From this perspective, variables with variables with VIP < 0.8, low weight in the loading plot or low coefficient. Moreover, between descriptors encoding the same information, those which performed better were chosen. It should be mentioned, however, that PLS, as a projection method, can tolerate intercorrelated variables, as compared to multiple linear regression. The predictive ability and robustness of the models was evaluated using cross-validation as an internal validation according to the seven-fold option of Simca-P. The sum of the squared differences between the measured response and the predicted value of the omitted data, defined as predicting residuals sum of squares (PRESS), is used to calculate the cross-validated correlation coefficient, Q^2^: Q^2^ = 1 − PRESS/SSY, with SSY representing the variation in Y after mean centering and scaling. Permutation tests (100 permutations) were applied by randomly re-ordering the response variables, and the newly derived R^2^ and Q^2^ were plotted against the degree of correlation between the permuted and the original data. The models were validated by external test sets, as already described for multiple linear regression. The root mean square error of prediction (RMSEP) was considered as an index of the predictability of the models. The applicability domain was defined using the double of the average value of the distance of observation to model Y (DModY) as the critical value.

## 4. Conclusions

In the present study, we combined bio-chromatographic retention factors, determined on IAM, has, and AGP stationary phases, with computed descriptors to develop MLR/PLS models for the rapid estimation of drug brain disposition. Our aim was to suggest a novel ‘hybrid’ modelling approach which combines high-throughput experimental accuracy with theoretical descriptors. At the same time, the well-investigated information content of the individual biomimetic properties [24,29,44,45] permitted a deeper insight into the underlying mechanisms of biological processes. In this sense, diverse aspects of drug disposition in the CNS were explored, including the modeling of experimental logK_p,uu,brain_, logBB_,_ f_u,brain_, and V_u,brain_ data, upon the application of two statistical techniques, considered as contributing in a complementary way. MLR led to simple models that are easy to use by the medicinal chemist, whereas PLS served to strengthen the MLR models and to further scrutinize the biological issues inherent in brain disposition measurements.

LogK_p,uu,brain_, as the outcome of contradictory factors related to permeability and tissue and plasma binding could not be efficiently modeled by biomimetic properties and lipophilicity, while computational descriptors alone were sufficient for model construction. Yet, PLS analysis revealed the greater effect of IAM, HSA, and AGP retention factors, as well as logD, if measured at pH 5.0, supporting the potential interference of the endo-lysosomic system in vitro brain penetration models. In regard to logBB, which measures permeability to CNS, traditional lipophilicity and IAM retention showed equal performance with positive contributions. However, in the cases of both logK_b_ and logV_ubrain_, which also depend on binding, IAM retention performed better than lipophilicity. More to the point, a lesser number of total descriptors was required in the corresponding PLS models, reflecting the higher information content of IAM retention. HSA retention was found to be important in logK_b_ modeling, while AGP retention influenced all brain disposition data, confirming the role of basicity as an essential CNS-drug-like characteristic. Polarity and hydrogen bond descriptors proved crucial in all models, with an opposite effect in regard to biomimetic properties and lipophilicity.

In view of the above findings, biomimetic properties are well-justified in modeling distinct brain disposition data as they reflect the major factors which govern biological processes, e.g., passive diffusion and binding. More to the point, IAM retention performs equally well or better than traditional lipophilicity. Hence, biomimetic chromatography can be suggested as a rapid and simple tool for the early evaluation of CNS drug candidates, permitting the construction of evidence-based ‘hybrid’ models and bridging the gap with in silico modeling, built solely on theoretical descriptors.

The ‘hybrid ‘complementary models constructed in this study can be further applied to, and validated with, a wider range of pharmaceutical compounds. Combined with relevant plasma protein binding models also based on biomimetic properties [45], they can serve as a sound basis for exploring the composite brain disposition end-points.

## Figures and Tables

**Figure 1 molecules-27-03668-f001:**
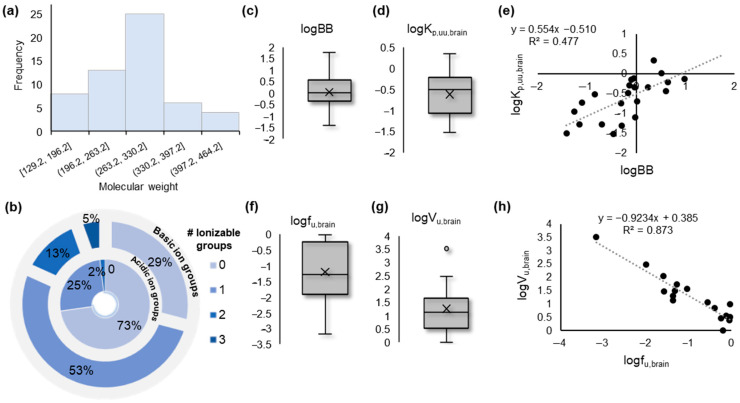
Overview of the data. (**a**) Histogram showing the distribution of the molecular weight of the investigated compounds; (**b**) Overlaid pie plots showing the number of ionizable groups (basic, outer layer; acidic, inner layer) present in the investigated compounds; (**c**) Box plot showing the distribution of the experimental logBB values collected from the literature; (**d**) Box plot showing the distribution of the experimental logK_p,uu,brain_ values collected from the literature; (**e**) Correlation scatter plot between logBB and logK_p,uu,brain_ values; (**f**) Box plot showing the distribution of the experimental logf_u,brain_ values collected from the literature; (**g**) Box plot showing the distribution of the experimental logV_u,brain_ values collected from the literature; (**h**) Correlation scatter plot between logf_u,brain_ and logV_u,brain_ values.

**Figure 2 molecules-27-03668-f002:**
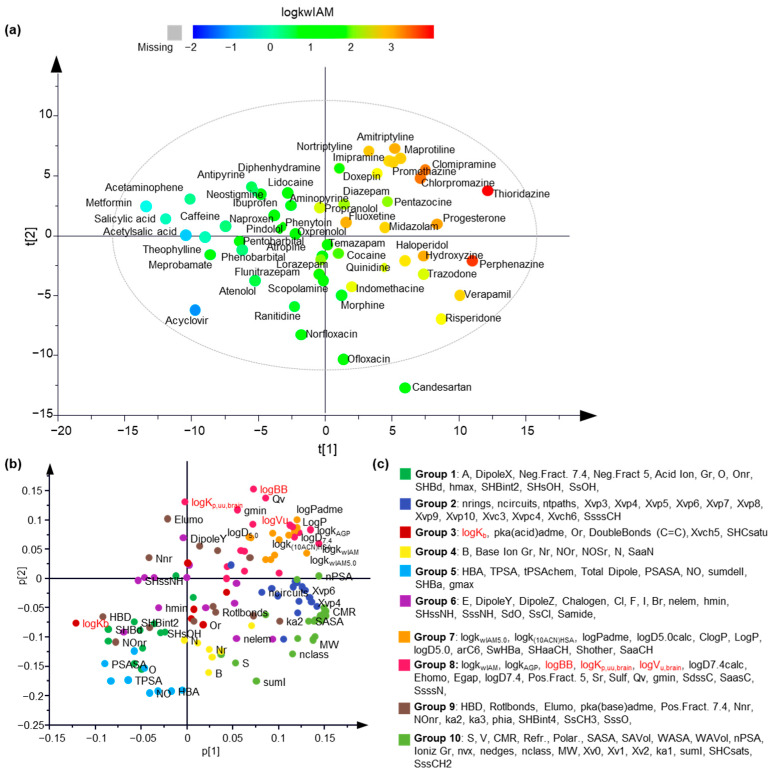
The unsupervised analysis provided a comprehensive overview of the data. (**a**) Scores plot of the first two principal components. The objects are colored according to a logk_wIAM_-based color scale; (**b**) Loadings plot of the first two principal components. The original variables are colored according to the hierarchical clustering grouping; (**c**) The grouping of the original variables based on hierarchical clustering.

**Figure 3 molecules-27-03668-f003:**
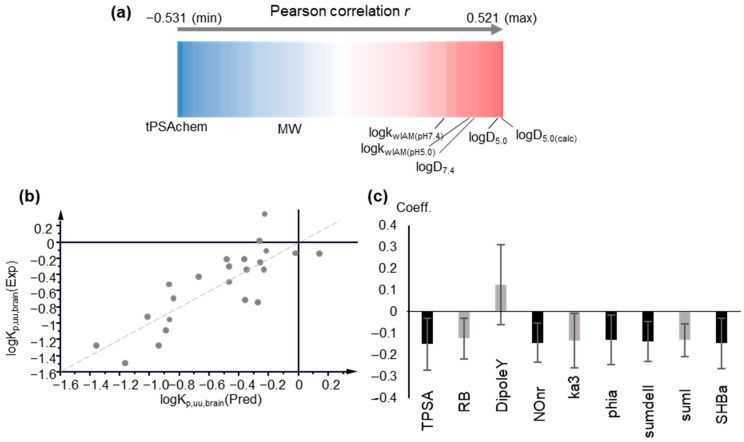
Modeling of logK_p,uu,brain._ (**a**) Pearson r correlation coefficients between logK_p,uu,brain_, physicochemical and molecular descriptors, and chromatographic data, depicted as a one-dimensional heat map; (**b**) Observed vs. predicted logK_p,uu,brain_ values plot based on PLS Model 4, including computational descriptors; (**c**) Coefficient plot of the original variables of PLS Model 4. Variables with VIP > 1 are highlighted in black.

**Figure 4 molecules-27-03668-f004:**
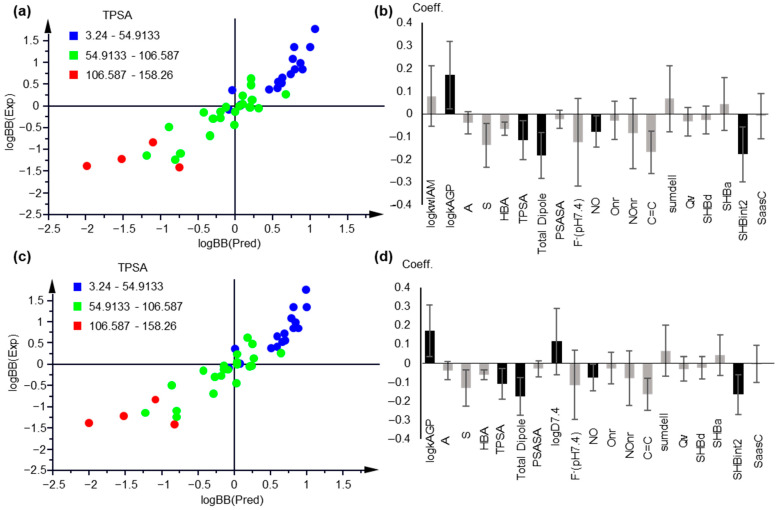
PLS modeling of logBB based on chromatographic and lipophilicity data. (**a**) Observed vs. predicted logBB values plot, based on PLS Model 5; (**b**) Coefficient plot of the original variables of PLS Model 5; (**c**) Observed vs. predicted logBB values plot, based on PLS Model 6; (**d**) Coefficient plot of the original variables of PLS Model 6. In (**a**,**c**), the compounds are colored based on their TPSA values. In (**b**,**d**), variables with VIP > 1 are highlighted with black.

**Figure 5 molecules-27-03668-f005:**
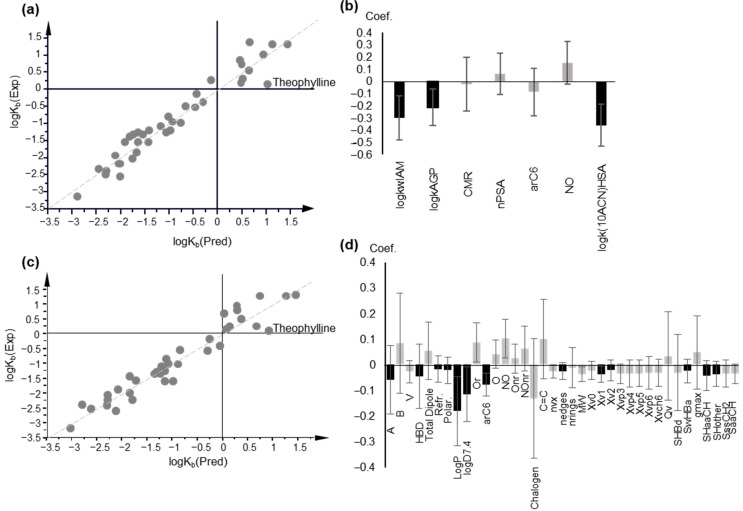
PLS modeling of logK_b_, based on chromatographic data and lipophilicity parameters. (**a**) Observed vs. predicted logK_b_ values plot based on PLS Model 8; (**b**) Coefficient plot of the original variables of PLS Model 8; (**c**) Plot of observed vs. predicted logKb values, based on PLS Model 9; (**d**) Coefficient plot of the original variables of PLS Model 9; In (**b**,**d**), variables with VIP > 1 are highlighted in black.

**Figure 6 molecules-27-03668-f006:**
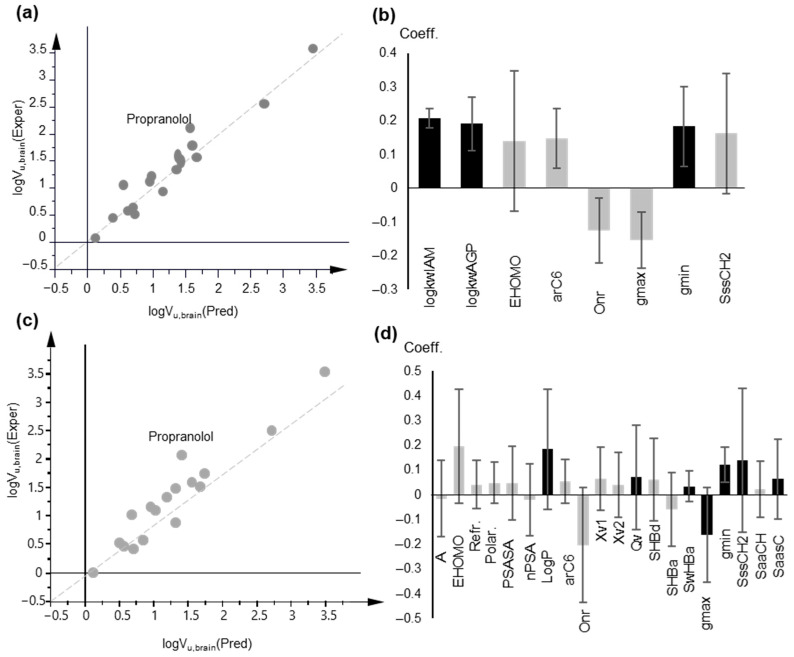
PLS modeling of logV_u,brain_ based on chromatographic data and lipophilicity parameters. (**a**) Observed vs. predicted logV_u,brain_ values plot based on PLS Model 10; (**b**) Coefficient plot of the original variables of PLS Model 10; (**c**) Observed vs. predicted logV_u,brain_ values plot based on PLS Model 11; (**d**) Coefficient plot of the original variables of PLS Model 11; In (**b**,**d**), variables with VIP > 1 are highlighted in black.

**Table 1 molecules-27-03668-t001:** Ranges of statistical values for training and external test sets used for validation of model 4 for logK_p_,_uu,brain_.

Validated PLS Model: 4 Different Test Sets with n = 6	R^2^_train_/Q^2^_train_	RMSEE	R^2^_test_	RMSEP
PLS Model 4 (based on computational descriptors)	0.687–0.831/0.597–0.736	0.212–0.276	0.212 *–0.824	0.229–0.407 *

R^2^_train_, coefficient of determination of the training sets; R^2^_test_, coefficient of determination of the test sets; Q^2^_train_, cross-validated coefficient of determination in PLS models; RMSEE, root mean square error of estimation; RMSEP, root mean square error of prediction; * statistics deteriorated due to the presence of acyclovir in the test set.

**Table 2 molecules-27-03668-t002:** Ranges of statistical values for training and external test sets for logBB validated models.

Validated MLR Models: 6 Different Test Sets with n = 7 or n = 6	R^2^_train_	s_train_	R^2^_test_	s_test_
Equation (1) (based on IAM retention)	0.626–0.705	0.388–0.499	0.583–0.932	0.167–0.669
Equation (2) (based on lipophilicity)	0.639–692	0.416–0.472	0.532–0.848	0.244–0.510
**Validated PLS Models: 5 Different Test Sets with** **n = 8 or n = 9**	**R^2^_train_/Q^2^_train_**	**RMSEE**	**R^2^_test_**	**RMSEP**
PLS Model 5 (based on IAM retention)	0.850–0.863/ 0.724–0.793	0.272–0.316	0.457 *–0.936	0.302–0.525 *
PLS Model 6 (based on lipophilicity)	0.850–0.865/ 0.725–0.791	0.272–0.328	0.600 **–0.919	0.317–0.516 **

R^2^_train_, coefficient of determination of the training sets; s_train_, standard deviation of the training sets; R^2^_test_, coefficient of determination of the test sets; s_test_, standard deviation of the test sets; Q^2^_train_, cross-validated coefficient of determination in PLS models; RMSEE, root mean square error of estimation; RMSEP, root mean square error of prediction. * Deteriorated statistics due to the presence of quinidine. ** Deteriorated statistics due to the presence of morphine.

**Table 3 molecules-27-03668-t003:** Ranges of statistical values for the training and external test sets for logK_b_ validated models.

Validated MLR Models: 5 Different Test Sets with n = 8 or n = 9	R^2^_train_	s_train_	R^2^_test_	s_test_
Equation (7) (IAM retention)	0.924–0.937	0.306–0.341	0.823–0.966	0.270–0.377
Equation (11) (lipophilicity)	0.848–0.890	0.403–0.462	0.797–0.955	0.310–0.664
**Validated PLS Models: 5 different test sets with n = 8 or = 9**	**R^2^_train/_Q^2^_train_**	**RMSEE**	**R^2^_test_**	**RMSEP**
PLS Model 8 (based on IAM retention)	0.919–0.946/ 0.851–0.927	0.286–0.353	0.822–0.988	0.277–0.519
PLS Model 9 (based on lipophilicity)	0.751–0.931/ 0.669–0.857	0.358–0.610	0.711–0.939	0.402–0.676

R^2^_train_, coefficient of determination of the training sets; s_train_, standard deviation of the MLR models on the training sets; R^2^_test_, coefficient of determination of the test sets; s_test_, standard deviation of the test sets; Q^2^_train_, cross-validated coefficient of determination for PLS models: RMSEE, root mean square error of estimation; RMSEP, root mean square error of prediction.

**Table 4 molecules-27-03668-t004:** Ranges of statistical values for the training and external test sets for models validated by logV_u,brain_.

Validated MLR Model: 3 Different Test Sets with n = 4–6	R^2^_train_	S_train_	R^2^_test_	s_test_
Equation (15) (IAM retention)	0.686–0.923	0.262–0.376	0.433 *–0.920	0.257–0.434
Equation (17) (lipophilicity)	0.526 *–0.852	0.364–0.485	0.437 *–0. 0.950	0.132–0.429
**Validated PLS Model: 2 different test sets with n = 8–9**	**R^2^_train_/Q^2^_train_**	**RMSEE**	**R^2^_test_**	**RMSEP**
PLS Model 10 (based on IAM retention)	0.927, 0.962/ 0.690, 0.922	0.160, 0.268	0.718, 0.956	0.316, 0.656
PLS Model 11 (based on lipophilicity)	0.747, 0.962/ 0.747, 0.880	0.160, 0.251	0.718, 0.782	0.650, 0.656

R^2^_train_, coefficient of determination of the training sets; s_train_, standard deviation of the MLR models on the training sets; R^2^_test_, coefficient of determination of the test sets; s_test_, standard deviation of the test sets; Q^2^_train_, cross-validated coefficient of determination in PLS models; RMSEE, root mean square error of estimation; RMSEP, root mean square error of prediction; * Deteriorated statistics due to the presence of metformin.

## Data Availability

All experimental and predicted brain disposition data are provided in the Appendix A. All experimental chromatographic and lipophilicity data are provided in the Appendix A.

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
