# Peer review of "Prediction Models for Brain Distribution of Drugs Based on Biomimetic Chromatographic Data"

_molecules, 2022, doi:10.3390/molecules27123668_

Round 1
Reviewer 1 Report
The manuscript entitled “Prediction models for brain distribution of drugs based on biomimetic chromatographic data” by Tsantili –Kakoulidou et al reported the good peace of work using modelling and in silico work for the model development and evaluating the properties. The author in the current studies investigated the biomimetic properties of 55 pharmaceutical compounds with varying classes. The prediction of the composite data with brain disposition components Kp,brain, fu,brain, and Vu,brain were carried out. Various descriptors were used and models were generated. Model construction was done using multiple linear regression and partial least square analysis. Based upon their findings, the authors suggested that biomimetic chromatography can be suggested as a rapid and simple tool for early evaluation of CNS drug candidates, permitting the construction of evidence based ‘hybrid’ models and bridging the gap with in silico modeling, based solely on theoretical descriptors.
I am happy to accept the article after revision.
In Figure 1a, histogram should be drawn again for more clarity.
Figure 2 should be set as above and below figures instead of parallel because plots are not readable.
Table 1-4, foot notes should be added for RMSEE, RMSEP, etc
Conclusion section should be rewritten.
Author Response
The manuscript entitled “Prediction models for brain distribution of drugs based on biomimetic chromatographic data” by Tsantili –Kakoulidou et al reported the good peace of work using modelling and in silico work for the model development and evaluating the properties. The author in the current studies investigated the biomimetic properties of 55 pharmaceutical compounds with varying classes. The prediction of the composite data with brain disposition components Kp,brain, fu,brain, and Vu,brain were carried out. Various descriptors were used and models were generated. Model construction was done using multiple linear regression and partial least square analysis. Based upon their findings, the authors suggested that biomimetic chromatography can be suggested as a rapid and simple tool for early evaluation of CNS drug candidates, permitting the construction of evidence based ‘hybrid’ models and bridging the gap with in silico modeling, based solely on theoretical descriptors.
I am happy to accept the article after revision.
We thank the reviewer for the comprehensive description of our work and the positive comments.
In Figure 1a, histogram should be drawn again for more clarity.
The histogram has now been updated; the x and y axes have been clearly annotated.
Figure 2 should be set as above and below figures instead of parallel because plots are not readable.
Figure 2 has now been revised according to reviewer’s recommendations. The font and size of the plots have also increased.
Table 1-4, foot notes should be added for RMSEE, RMSEP, etc
We thank the reviewer for noticing this neglect, the information has now been added (lines 245-247).
Conclusion section should be rewritten.
The Conclusion section has now been rewritten, to highlight better the novelty and future perspectives of the study (lines 625 -662):
Reviewer 2 Report
All the cited references are well selected, although they concern only a certain, narrow area of drug penetration into the CNS. The research was carried out on a small group of cases. Additionally, this group is limited by data availability. The studied groups of cases are very small in some analyzes.
Author Response
Reviewer #2
All the cited references are well selected, although they concern only a certain, narrow area of drug penetration into the CNS. The research was carried out on a small group of cases. Additionally, this group is limited by data availability. The studied groups of cases are very small in some analyzes.
The data set is indeed rather limited, since we had to combine availability of both biomimetic measurements and drug disposition data. The aim was to show the suitability of bio- chromatography to evaluate CNS drug candidates. Being aware of the limited data and in order to support the results we used two statistical techniques which can be considered to be complementary while we examined in detail model applicability domain, as also mentioned in the aim of the study. It is our intention to expand our studies to include wider data sets of pharmaceutical compounds as also stated in the revised. These issues have been added in the revised manuscript. (lines 119-121 and revised conclusions)
Reviewer 3 Report
The manuscript entitled “Prediction models for brain distribution of drugs based on biomimetic chromatographic data” aims to investigate the performance of biomimetic properties in combination with molecular descriptors for the development of ‘hybrid’ models with experimental evidence for the prediction of the ratio between the unbound drug concentrations in the interstitial fluid of the brain to the corresponding plasma concentration. In-house retention factors, determined for a number of pharmaceutical compounds on IAM, HSA and AGP stationary phases, were used together with physicochemical/molecular descriptors. Models were constructed applying multiple linear regression and partial least squares analysis. Attention was given to model validation with respect to their robustness and applicability domain to offset the drawback that rather limited datasets were analyzed. This topic is very important since the development of novel compounds targeting the CNS is becoming more and more essential, and the blood-brain barrier is the most important obstacle to drug delivery in the brain. The manuscript is well prepared. I have some small concerns. First of all, the novelty of the proposed method should be emphasized more. Also, a future perspective on this kind of methods advancement should be offered.
Some additional comments:
“demanding and time-consuming and inefficient for “ line 57
“included in the Materials and Methods section. “ line 64
“been reported in the literature, most of them” line 69
“errors in the estimation of biological endpoints, in particular” line 76
“the other hand, user-friendly chromatographic techniques may“ line 78
“reported in the literature“ line 90
“On the other hand, protein-based stationary phases“ line 93
“were constructed by applying multiple linear regression (MLR) and partial least” line 118
“components show a uniform distribution of the data in all four quartiles” line 155
“The unsupervised analysis provided a” line 168
“into the training set and test set, using 4 different training/test sets “line 225
“have a lower influence in the model with” line 275
“indicates that a higher hydrogen bond a“ line 335
“the relationship was obtained although with a lower correlation “line 351
“into the training set and test set, using 5 different test sets (Table 3). “ line 368
“have a negative sign” line 381
“for the unbound volume of distribution, using” line 427
“A one-component PLS model, based on IAM retention, was obtained ” line 460
“have a negative effect, and the opposite is true for non-polar” line 465
“the mobile phase were used. Two buffer pH values were used” line 513
“variables that summarize the information“ line 595
“was also considered for test sets selection. Partial least squares analysis (PLS)” line 597
Author Response
Reviewer #3
The manuscript entitled “Prediction models for brain distribution of drugs based on biomimetic chromatographic data” aims to investigate the performance of biomimetic properties in combination with molecular descriptors for the development of ‘hybrid’ models with experimental evidence for the prediction of the ratio between the unbound drug concentrations in the interstitial fluid of the brain to the corresponding plasma concentration. In-house retention factors, determined for a number of pharmaceutical compounds on IAM, HSA and AGP stationary phases, were used together with physicochemical/molecular descriptors. Models were constructed applying multiple linear regression and partial least squares analysis. Attention was given to model validation with respect to their robustness and applicability domain to offset the drawback that rather limited datasets were analyzed. This topic is very important since the development of novel compounds targeting the CNS is becoming more and more essential, and the blood-brain barrier is the most important obstacle to drug delivery in the brain. The manuscript is well prepared.
We thank the reviewer for the thorough description of our study and the positive comments.
I have some small concerns. First of all, the novelty of the proposed method should be emphasized more.
We thank the reviewer for pointing this out. We have now revised the Introduction and the Conclusions sections accordingly, clearly stating the novelty of our approach:
lines … “109-114”, “121-124..”
lines…. “628-638”
Also, a future perspective on this kind of methods advancement should be offered.
This is a very good point. A relevant sentence has been added in the Conclusions section, lines
“662-665”
Some additional comments:
“demanding and time-consuming and inefficient for “ line 57 (
“included in the Materials and Methods section. “ line 64
“been reported in the literature, most of them” line 69
“errors in the estimation of biological endpoints, in particular” line 76
“the other hand, user-friendly chromatographic techniques may“ line 78
“reported in the literature“ line 90
“On the other hand, protein-based stationary phases“ line 93
“were constructed by applying multiple linear regression (MLR) and partial least” line 118 (120)
“components show a uniform distribution of the data in all four quartiles” line 155
“The unsupervised analysis provided a” line 168 (173)
“into the training set and test set, using 4 different training/test sets “line 225 (230)
“have a lower influence in the model with” line 275 (284)
“indicates that a higher hydrogen bond a“ line 335 (344)
“the relationship was obtained although with a lower correlation “line 351 (360)
“into the training set and test set, using 5 different test sets (Table 3). “ line 368 (377)
“have a negative sign” line 381 (390)
“for the unbound volume of distribution, using” line 427 (436)
“A one-component PLS model, based on IAM retention, was obtained ” line 460 (469)
“have a negative effect, and the opposite is true for non-polar” line 465 (474)
“the mobile phase were used. Two buffer pH values were used” line 513 (522)
“variables that summarize the information“ line 595 (605)
“was also considered for test sets selection. Partial least squares analysis (PLS)” line 597 (607)
We thank the reviewer for noticing these typos/grammatical inaccuracies. The text has been formatted accordingly and the revisions are highlighted with “Track changes”. In parentheses are the corresponding lines in the revised manuscropt Regarding the line 155 (160), the verb is in singular form since the subject is “the score plot”. Sentence at line 513 (522) has been revised and PBS as the buffer has been added.